# Application of Sensor Path Weighting RAPID Algorithm on Pitting Corrosion Monitoring of Aluminum Plate

**DOI:** 10.3390/ma15113887

**Published:** 2022-05-30

**Authors:** Duo Xu, Weifang Zhang, Lu Han, Xuerong Liu, Weiwei Hu

**Affiliations:** 1School of Reliability and Systems Engineering, Beihang University, 37 Xueyuan Rd., Haidian District, Beijing 100191, China; xuduo@buaa.edu.cn (D.X.); 08590@buaa.edu.cn (W.Z.); by2014110@buaa.edu.cn (L.H.); 2School of Aeronautic Science and Engineering, Beihang University, 37 Xueyuan Rd., Haidian District, Beijing 100191, China; liuxuerong@buaa.edu.cn

**Keywords:** pitting corrosion, PZTs, SPW-RAPID, SDC, RMSE, *E*1

## Abstract

Aluminum alloy is widely used in aerospace structures. However, it often suffers from a harsh corrosion environment, resulting in different damage such as pitting corrosion, which leads to a reduction in the service life of aerospace structures. In the present study, the pitting corrosion with a radius of 1 mm and a depth of 0.6 mm was manufactured using hydrofluoric (HF) acid on a 2024-T3 aluminum alloy plate (400 mm × 400 mm × 2 mm) to simulate the corrosion state of equipment. A signal acquisition system with a square sensor network of 12 piezoelectric transducers (PZTs) was established. The sensor path weighting reconstruction algorithm for the probabilistic inspection of defects (SPW-RAPID) is proposed based on corrosion damage characteristic parameters including signal correlation coefficient (SDC), root mean squared error (RMSE), and signal energy damage index (*E*1) to explore the monitoring efficacy of pitting corrosion. The sensor path weight *w*, which is the product of value coefficient *a* and impact factor *l*, is established to modify the corrosion damage characteristic parameters. The results indicate that the SPW-RAPID algorithm can improve the accuracy and clarity of image reconstruction results based on SDC, RMSE and *E*1, which can locate the pitting corrosion with a radius of 1 mm and a depth of 0.6 mm, and the positioning error is controlled within 0.1 mm. The research work may provide an available way to monitor tiny corrosion damage on an aluminum alloy structure.

## 1. Introduction

Aluminum alloy is widely used in aerospace structures because of its excellent properties such as low density, high strength, good mechanical properties and low cost [1]. Meanwhile, pitting damage is a kind of pervasive, hazardous and insidious corrosion damage that can easily emerge on a boldly exposed surface of aluminum alloy [2]. Exposure to acidic agents such as chlorides and sulfides will cause corrosion of metals, which brings about tiny substances peeling off from the surface. Importantly, once the corroded metals are subjected to mechanical stress, the pitting corrosion will finally cause other damage, such as fatigue cracks and deeper corrosion [3,4]. This progress is of large concern with respect to aging aircraft, because maintained aging aircrafts with pitting corrosion could lead to disastrous air accidents when exposed to an aggressive environment [2,3]. So, monitoring pitting corrosion initiation early provides valuable data for structure health monitoring and aids in reducing the cost of aircraft maintenance.

Existing pitting corrosion identification methods can be roughly divided into visual inspection and mechanical methods. Current visual tests include the application of optical instruments such as optical microscopy (OM), scanning electron microscopy (SEM), scanning acoustic microscopy (SAM), energy dispersive X-ray spectroscopy (EDS), and so on. In this way, the outline, size and depth of pitting can be directly observed. For instance, since 1998, the OM, SEM and EDS have been used to observe the initiation and development trend of pitting corrosion on a 2024-T3 aluminum alloy plate; finally, it is founded that combined with this equipment, the location and range of pitting corrosion can be determined [1]. In 1999, Frantziskonis et al. adopted a white light interference microscope and inspected pitting corrosion on 2024-T3 aluminum alloy as well [5]. In 2001, a study demonstrated that image analysis based on light microscopy can be effectively used as a tool to quantitatively characterize the morphology of pitting corrosion [6]. Until now, some scholars evaluated the intensity and depth of corrosion pits on aluminum alloy 7050 using SAM, and the results were cross-checked by OM [7].

Traditional mechanical methods are composed of passive sensing using acoustic emission (AE) and active sensing using guided wave. AE generates elastic waves by piezoelectric transducers (PZTs) to detect the rapid energy release inside the structure, which can be used to measure the existence of damage, and it has already been proven as a helpful tool for early-stage detection of pitting corrosion [7,8,9]. In 1995, Mazille was the first one to apply AE to detect and study the development of pitting corrosion on AISI 316L austenitic stainless steel [8]. Trdan et al. applied AE during accelerated electrochemical potentiostatic tests and found it a reliable method for the early detection of pitting corrosion of investigated Al alloy and for general monitoring of corrosion activity [7].

Active sensing is a method that can generate and sense ultrasonic guided waves with a number of transducers such as embedded or surface-bonded PZTs and laser doppler vibrometer [2,10]. In recent studies, due to the ability of detecting damages across a wide area with fewer sensors, active sensing is widely used in health monitoring in aluminum alloy plate [11]. Lamb wave tomography based on PZTs can apply imaging methods to locate pitting damage, such as the phased array-based imaging method, spatial filter-based imaging method and reconstruction algorithm for probabilistic inspection of defects (RAPID). For example, a combination of dispersion compensation and deconvolution in phased-addition algorithm is used; therefore, simulated pitting corrosion could be well detected within the dynamic range of the array measurement [12]. Some studies have revealed that a method using scanning laser Doppler vibrometer (SLDV) can detect a cluster of pits which are arranged in 3 × 3 × 3 array and have the diameter of 2 mm and the interval of 2 mm in aluminum plate, but most pits except the one at the center of cluster can be identified [13,14]. Recently, Cao et al. proposed the NI, which can link the degree of nonlinearity to the size of pitting damage, and both the PZTs network and RAPID algorithm are used to visualize pitting damage, but only an approximate range of pitting corrosion can be located [15].

However, the limitations of visual inspection include big subjective factors, disturbing human interference, and a time-consuming and laborious process. The most inconvenient is that it is unable to observe damage in real time. Although passive acting using AE can determine whether pitting exists in the structure in real time, it cannot accurately determine the specific position, shape and size of pitting damage. Laser doppler vibrometers are limited in its heavy weight, which is unrealistic to carry on the aircraft for use. To overcome these problems, PZTs are a good choice, which are light-weight, inexpensive, highly efficient and flexible [10]. In addition, different shapes of sensor network with PZTs can be used to monitor damage in specified areas [16].

In this paper, hydrofluoric (HF) acid is used to produce pitting corrosion with a radius of 1 mm and depth of 0.6 mm on a 2024-T3 aluminum plate. Both the network with 12 piezoelectric sensors (PZTs) and ScanGenie-II integrated structural health monitoring scanning system is built. The proposed SPW-RAPID algorithm based on different corrosion damage characteristic parameters is used to locate the pitting corrosion and characterize its outline and size. Finally, the pitting corrosion with a radius of 1 mm and a depth of 0.6 mm on a 2024-T3 aluminum plate can be successfully detected.

## 2. Tomography Algorithm

### 2.1. Conventional RAPID Algorithm

The RAPID algorithm is identified as a critical probabilistic method for reconstructing damage images, which can characterize the difference between the signals interacting with damage and the baseline signals without damage [17]. The signal comparison is mostly on account of damage characteristic parameters, which can sense the subtle change in transmitted signal and evaluate the damage severity of the detected area [17], such as signal difference coefficient (SDC), root mean squared error (RMSE), damage index based on signal energy *E*1, *E*2, *E*3 and normalized correlation moment (NCM). All of them can be applied to the RAPID algorithm to identify the signal difference, which are calculated as follows respectively:Signal Difference Coefficient [18]:
(1)$$SDC_{\mathrm{ij}}=1-\left|\frac{\sum_{n=1}^{N}\left[x_{\mathrm{ij}}\left(n\right)-\mu \right]\left[y_{\mathrm{ij}}\left(n\right)-\mu \right]}{\sqrt{\sum_{n=1}^{N}{\left[x_{\mathrm{ij}}\left(n\right)-\mu \right]}^{2}\sum_{n=1}^{N}{\left[y_{\mathrm{ij}}\left(n\right)-\mu \right]}^{2}}}\right|$$Root Mean Squared Error [19]:
(2)$$RMSE_{\mathrm{ij}}=\sqrt{\frac{1}{N}\sum_{n=1}^{N}{\left[x_{\mathrm{ij}}\left(n\right)-y_{\mathrm{ij}}\left(n\right)\right]}^{2}},$$Damage index based on signal energy *E*1 [20]:
(3)$$E1_{\mathrm{ij}}=\frac{\sum_{n=1}^{N}{\left|x_{\mathrm{ij}}\left(n\right)\right|}^{2}}{\sum_{n=1}^{N}{\left|y_{\mathrm{ij}}\left(n\right)\right|}^{2}},$$Damage index based on signal energy *E*2 [21]:
(4)$$E2_{\mathrm{ij}}=\frac{\sum_{n=1}^{N}{\left|x_{\mathrm{ij}}\left(n\right)-y_{\mathrm{ij}}\left(n\right)\right|}^{2}}{\sum_{n=1}^{N}{\left|y_{\mathrm{ij}}\left(n\right)\right|}^{2}},$$Damage index based on signal energy *E*3 [22]:
(5)$$E3_{\mathrm{ij}}=\frac{\sum_{n=1}^{N}{\left|x_{\mathrm{ij}}\left(n\right)\right|}^{2}-\sum_{n=1}^{N}{\left|y_{\mathrm{ij}}\left(n\right)\right|}^{2}}{\sum_{n=1}^{N}{\left|y_{\mathrm{ij}}\left(n\right)\right|}^{2}},$$Normalized Correlation Moment [23]:
(6)$$NCM_{\mathrm{ij}}=\frac{\sum_{n=1}^{N}n^{k}\left|r_{xy}\left(n\right)\right|-\sum_{n=1}^{N}n^{k}\left|r_{xx}\left(n\right)\right|}{\sum_{n=1}^{N}n^{k}\left|r_{xx}\left(n\right)\right|},$$
(7)$$r_{xy}\left(n\right)=\sum_{n=1}^{N}x_{\mathrm{ij}}\left(n\right)y_{\mathrm{ij}}\left(N-n\right),$$
(8)$$r_{xx}\left(n\right)=\sum_{n=1}^{N}x_{\mathrm{ij}}\left(n\right)x_{\mathrm{ij}}\left(N-n\right).$$

Here, μ is the mean of the corresponding signal, $x_{\mathrm{ij}}\left(n\right)$ is the health signal from the transmitter *i* and receiver *j* sensor pair, and $y_{\mathrm{ij}}\left(n\right)$ is the damage signal from the transmitter *i* and receiver *j* sensor pair. *N* is the sampling length. $r_{xy}\left(n\right)$ is the cross-correlation function between health signal and damage signal. $r_{xx}\left(n\right)$ is the auto-correlation function of health signal. *k* is the order of statistical moment, and its value can be any positive number; when 0.01 is taken, the damage sensitivity is the highest [23].

After the damage index values of all sensor paths are calculated, the second step of this algorithm is image reconstruction. The shape factor β can measure the size of the elliptical distribution, which is usually greater than 1.0 [24]. The spatial distribution function $s_{ij}\left(x,y\right)$ is expressed as [17]: (9)$$\left\{\begin{array}{ll} s_{ij}\left(x,y\right)=\frac{\beta -R_{ij}\left(x,y\right)}{\beta -1} & \;\mathrm{for}\beta >R_{ij}\left(x,y\right)\\ s_{ij}\left(x,y\right)=0 & \;\mathrm{for}\beta \leq R_{ij}\left(x,y\right) \end{array}\right.,$$
where (x,y) are the coordinates of each point in a sensor array.

As shown in Figure 1, $R_{ij}\left(x,y\right)$ is the ratio of the sum of $\vec{D_{Ai}}$ and $\vec{D_{iB}}$ to the distance between transmitter–receiver pairs $\vec{D_{i}}$, which is expressed as [17]: (10)$$R_{ij}=\frac{\sqrt{{\left(x_{i}-x\right)}^{2}+{\left(y_{i}-y\right)}^{2}}+\sqrt{{\left(x_{j}-x\right)}^{2}+{\left(y_{j}-y\right)}^{2}}}{\sqrt{{\left(x_{j}-x_{i}\right)}^{2}+{\left(y_{j}-y_{i}\right)}^{2}}},$$

Finally, $D_{ij}$ is defined as the damage characteristic parameters at each pixel, which represents the value of SDC, RMSE, *E*1, *E*2, *E*3 and NCM. So, the damage probability of each point (x,y) of the detected area is calculated as follows [17]: (11)$$P\left(x,y\right)=\sum_{i=1}^{N}\sum_{j=1,j\neq i}^{N}DI_{ij}s_{ij}\left(x,y\right),$$

In the following study, we try to characterize the behavior of SDC, RMSE, E1, E2, E3 and NCM on monitoring pitting corrosion with the RAPID algorithm. In addition, in this work, we tried to distinguish a clear cognition to the effect of various corrosion damage characteristic parameters on image reconstruction so as to determine the best monitoring method.

### 2.2. Sensor Path Weighting RAPID Algorithm

Generally speaking, the S0 mode and A0 mode have a similar effect on identifying multiple structural damage. However, the size of pitting corrosion in this paper is too tiny, and corrosion damage leads to the change of aluminum plate thickness. As we know, the Lamb wave A0 mode is sensitive to thickness variations, which is superior to that of S0 mode [25]. Another essential property is that the A0 mode outperforms the S0 mode with a shorter wavelength and larger signal magnitude at relatively low frequencies (such as 110 kHz), which is beneficial to detect tiny damage [26]. So, the A0 mode is selected to detect the signal differences caused by pitting corrosion.

In the process of extracting damage feature information from the signal, two considerable factors are used to study the effect on the results, and both of them are related to the status of the sensor path. The first factor is whether the sensor path directly passes through the corrosion damage. Because when encountering damage, the ultrasonic signals will obtain different waves, such as reflected waves, diffracted waves, energy-attenuated waves and mode-converted waves [27]. Subsequently, the amplitude and phase of signals are changed. So, it finally can represent whether a Lamb wave propagating on the sensor path carries effective damage feature information or not. On those paths not directly crossing through the corrosion part, the damage cannot interact with signals, which has a subtle effect on the A0 mode. Simultaneously, with the increase of the distance between sensor path and corrosion damage, the impact of damage on A0 mode gradually decreases. Therefore, the linear distance from sensor paths to pitting corrosion is established as $d_{0}$, which can evaluate how sensor paths are affected by corrosion damage.

The second factor is the length of sensor path, which means the distance between sensor pairs. As expected, if sensor paths meet the condition that different Lamb wave modes can separate completely, it is convenient to obtain the damage feature information in signals. On the contrary, short sensor paths cannot provide sufficient propagation distance for signals, so the crosstalk in front, S0 mode and A0 mode will overlap with each other, making it difficult to obtain the important part of scattered signals [28]. However, the existing RAPID algorithm does not take into account these two factors. Based on this problem, the algorithm needs to be modified.

The sensor path weighting RAPID (SPW-RAPID) algorithm is proposed on the basis of the RAPID algorithm. The sensor path weight from the transmitter *i* and receiver *j* sensor pair is established as $w_{ij}$, which can modify and optimize the values of corrosion damage characteristic parameters. In addition, the value coefficient of sensor path from transmitter *i* to receiver *j* is proposed as $a_{ij}$, which can describe how the sensor path is affected by pitting corrosion based on the first factor. The impact factor of the sensor path from the transmitter *i* to receiver *j* is put forward as $l_{ij}$, which can evaluate the behavior of sensor path length to pitting corrosion identification based on the second factor. Finally, the sensor path is given a certain weight, which is defined as the product of value coefficient and impact factor on each sensor path, which is expressed as follows: (12)$$w_{ij}=a_{ij}\times l_{ij},$$
where *i* represents the excitation sensor, and *j* represents the receiving sensor.

First of all, the value coefficient $a_{ij}$ needs to be determined. As we know, the traditional RAPID algorithm can roughly locate the pitting corrosion in the detected area and obtain the original pitting location $\left(x_{0},y_{0}\right)$. A rectangular coordinate system is established based on the sensor network. Then, the linear expression of each sensor path in the coordinate can be obtained. According to the original pitting corrosion location $\left(x_{0},y_{0}\right)$, the perpendicular distance from pitting corrosion to any sensor path can be calculated, as shown below: (13)$$d_{0}=\frac{\left|Ax_{i}+By_{j}+C\right|}{\sqrt{A^{2}+B^{2}}},$$
where *A*, *B* and *C* represent the coefficients of the linear equation in a rectangular coordinate system, $x_{0}$ represents the horizontal ordinate of pitting position, and $y_{0}$ represents the longitudinal ordinate of the pitting position.

Secondly, the sensor path value is established as *f*, which describes the value level of sensor paths according to $d_{0}$ and finally defines the $a_{ij}$. The range of $a_{ij}$ is 0 to 1, where 1 represents the highest value and 0 represents no value. For instance, when $d_{0}$ is equal to 0, the sensor path directly passes through the damage, where the signals can carry the most useful damage feature information, so its value is very high and $a_{ij}$ can be assigned as 1. With the increase of $d_{0}$, the damage is gradually away from the path. So, the damage feature information carried by signals reduces correspondingly, which devalues the sensor path and decreases the value of $a_{ij}$. Additionally, those paths with a value coefficient of 0 are mostly located at the boundary of the sensor network. There is little important damage feature information in signals on the boundary path, which are vulnerable to plate edge reflection [29].

The third step is to determine the impact factor, whose range is 0 to 1 as well, and 1 represents that the influence of sensor path is great. Instead, 0 represents that there is no influence, because it is necessary to set a certain distance between sensors in order to extract an integral A0 mode. The distance between sensor pairs needs to satisfy as follows [30]: (14)$$\frac{L}{c_{A0}}-\frac{L}{c_{S0}}>\frac{n}{f_{0}},$$
where $c_{A0}$, $c_{S0}$ is the group velocity of the A0 mode and S0 mode, respectively; $f_{0}$ is the central frequency of the excitation signal; *n* indicates the number of cycles of the excitation signal; *L* is the distance between sensor pairs.

The $l_{ij}$ from transmitter *i* to receiver *j* is expressed as: (15)$$l_{ij}=\left\{\begin{array}{ll} 1 & \;\mathrm{for}L\geq m\;\mathrm{cm}\\ 0.2 & \;\mathrm{for}L<m\;\mathrm{cm} \end{array}\right.,$$

Finally, the weight on each sensor path and the original corrosion damage characteristic parameters are multiplied to obtain the new corrosion damage characteristic parameters. The spatial distribution function $S_{ij}$ and the damage probability P(x,y) are calculated as usual.

In order to evaluate the quality of imaging results, the accuracy and precision are put forward. Accuracy is the degree of closeness of barycentric coordinates between real damage and predicted damage. Distance Root Mean Squared (DRMS) is a good indicator to measure the distance between two points, as shown in Equation (Equation 16). The bigger the DRMS value is, the larger the localization error is. Meanwhile, precision is the degree of imaging clarity, which can be obtained by enlarging the images for comparison.
(16)$$DRMS=\sqrt{\sigma_{x}^{2}+\sigma_{y}^{2}},$$
where $\sigma_{x}$ represents the deviation of horizontal ordinate between real damage and predicted damage in a sensor network, and $\sigma_{y}$ represents the deviation of longitudinal ordinate between real damage and predicted damage in a sensor network.

## 3. Experimental Procedure

### 3.1. Specimen Design

The specimen is made of aluminum 2024-T3 with the dimension of 400 mm × 400 mm × 2 mm, as shown in Figure 2a. The material properties of the specimen are listed in Table 1. A square sensor layout with 12 piezoelectric transducers is designed as shown in Figure 2b. The pitch–catch configuration is adopted. So, there are 66 effective sensing paths, as shown in Figure 2c.

### 3.2. Monitoring System Setup

The acquisition experiment of Lamb wave signals was conducted by a ScanGenie-II integrated structural health monitoring scanning system produced by Acellent Technologies. The main technical indicators of the setup are listed in Table 2. The monitoring system consists of the testing specimen, piezoelectric transducers, ScanGenie-II piezoelectric monitoring device, signal generator, digital acquisition software and connecting box, as shown in Figure 3. This system equipped with a PZTs array is superior to acquire signals, because it can achieve multi-site and multi-frequency monitoring scanning of a large area at once.

From the dispersion equations, the relationship between group velocity and the product of frequency and specimen thickness is shown in Figure 4. As we can see, Lamb wave signals contain two or more modes at frequency above 2000 kHz. Wave modes at higher frequencies are inappropriate to detect the damage owing to its high energy attenuation during large distance propagation. To figure out the A0 mode and S0 mode completely and reduce the attenuation of wave mode, the frequency less than 2000 kHz is suitable to be selected. So, the excitation frequency of 110 kHz is selected in this paper.

The excitation signal is a five-peak sine tone burst, and the expression is expressed as follows [31]: (17)$$u\left(t\right)=A\left[H\left(t\right)-H\left(t-\frac{N}{f_{c}}\right)\right]\times \left(1-\cos\frac{2\pi f_{c}t}{N}\right)\sin2\pi f_{c}t,$$
where *A* is the signal amplitude, *N* is the number of crests, $f_{c}$ is the central frequency of the signal, and H(t) is the Heaviside step function. The sampling rate is 24 MSPS, and the sampling length is 10,000 points.

### 3.3. Pitting Corrosion Manufacture

After the preparation of experiment setup, simulated pitting corrosion with a radius of 1 mm and the depth of 0.6 mm is manufactured artificially. A series of corrosion devices include glass adhesive, hydrofluoric (HF) acid, glue dropper and medical syringe. The corrosion process is successively shown in Figure 5. HF acid provided by Hengxing Reagents was used to make artificial pitting corrosion on the specimen. The chemical compositions of HF acid are shown in Table 3.

An optimized method is proposed to directionally make an artificial pitting defect and ensure the accuracy of simulated manufacturing. Firstly, we apply a circular glass adhesive layer with the diameter of 1 cm and thickness of 5 mm in pre-etching area. Before the adhesive is fully cured, a medical syringe with a needle diameter of 1 mm is used to drill a small hole penetrating the adhesive layer in the pre-etching area. The simulated hole can be formed until the adhesive is completely cured. Simultaneously, the corrosive solution is prepared by HF acid and $H_{2}O$ with the ratio of 1:2. Subsequently, add the corrosive solution into the hole using a medical syringe. During the course of corrosion, replace the solution every 15 min until the corrosion process is done. The pitting corrosion is controlled by controlling the contact time between corrosive solution and specimen. As a result of this etching of an aluminum plate, a single pitting with a radius 1.0 ± 0.01 mm and a depth 0.60 ± 0.01 mm is obtained, as shown in Figure 5c,d). The tests are carried out three times to reduce error. At the beginning and the end of each test, the health signal and damage signal are collected separately by the device for subsequent signal processing.

## 4. Discussion and Results

Figure 6 clearly illustrates the simulated images using the conventional RAPID algorithm with different parameters. The red part where the damage probability is high represents that the pitting corrosion exists here. On the contrary, the blue part represents there is no pitting corrosion, and the damage probability here is low. It can be observed that images of SDC, RMSE and E1 exhibit the same as real corrosion, as shown in Figure 6a–c, respectively. They all image the pitting corrosion in the middle, and the size of imaging results is also similar to the reality. However, as shown in Figure 6c–e, the images of E2, E3 and NCM are thoroughly not in line with real damage. Some inefficiency still exists because there is another red artifact appearing in the image of RMSE. Although there is no artifact in the image of SDC and E1, there are many disturbing points all over the images, and the imaging clarity is low. The reason of poor imaging results is due to the small size of pitting corrosion, resulting in subtle signal differences, and the characteristic parameters cannot accurately distinguish those subtle differences.

On the basis of the method in Section 2.2, the imaging accuracy and clarity of 6 characteristic parameters are verified by DRMS and enlarged images. The barycentric coordinates of pitting corrosion are listed in Table 4. As we can see, the DRMS of SDC, RMSE and E1 are 0.141, 0.224, and 0.200 respectively, and the minimum deviation is only 0.200 mm. So, their localization error is small, which can be neglected. However, the localization error of E2, E3 and NCM is quite big, because the DRMS is up to 60.000 mm. The results revealed that only SDC, RMSE, and E1 with the RAPID algorithm can strongly interact with pitting corrosion and can roughly locate the damage position.

Subsequently, the simulated images were enlarged to evaluate the imaging clarity. Figure 7 shows the zoom-in view of simulated pitting based on SDC, RMSE and E1, respectively. All of them can roughly show the outline of pitting corrosion. The clearest one is E1, which contains no artifacts and characterizes the damage that is close to the real size. The images of SDC and RMSE take second place, because the red part is slightly smaller than the real state. All in all, the imaging quality needs to be improved.

Therefore, the SPW-RAPID algorithm in Section 2.2 is used to optimize the simulated images. Based on the sensor network in Figure 2c, a rectangular coordinate system is built up, and sensor 10 is taken as the grid origin, as shown in Figure 8. After calculation, the distances between pitting corrosion and sensor paths refer to eight different lengths as 0 cm, 2.5 cm, 3 cm, 4.24 cm, 5.69 cm, 6.71 cm, 8.49 cm and 9 cm, respectively. As listed in Table 4, there are six sensor paths with a distance of 0 cm defined as very high value, so the $a_{ij}$ is 1. All of them go directly through the pitting corrosion, such as paths 1–7. The paths with a distance of 2.5 cm that can be greatly affected by damage are defined as good value, such as paths 1–6, so the $a_{ij}$ is 0.5. Those with distances ranging from 3.00 to 8.49 cm are defined as general value, and the $a_{ij}$ is 0.2. The number of paths at the boundary of the sensor network is 24. On account of edge reflection, the $a_{ij}$ is defined as 0. Finally, the value and value coefficient are obtained as shown in Table 5.

Next, the impact factor needs to be determined. In this sensor network, if the length of the sensor path is less than 18.8 cm, the crosstalk in front and wave modes behind will overlap with each other. For instance, as shown in Figure 9a, the S0 mode does not appear at all on sensor paths 3–5, because it is covered by the crosstalk and A0 mode absolutely. Meanwhile, the A0 mode only appears to have four peaks, and the amplitude reduces relatively small compared to that of sensor paths 6–12. All in all, there are 44 paths achieving this condition in the sensor network, whose A0 modes only carry incomplete damage feature information, so they have little effect on pitting imaging; their impact factor is defined as 0.2.

However, if the sensor path length is above or equal to 18.8 cm, the crosstalk ahead, S0 mode and A0 mode can be completely separated. For example, the wave modes on sensor paths 6–12 have a complete shape, as shown in Figure 9b. It is discovered that the amplitude of the guided waves decreases with the increase of corrosion level, so the amplitude of the A0 mode in the damage state is lower than that in the health state. In summary, when the length of the sensor path is above or equal to 18.8 cm, the impact factor is defined as 1.

Figure 10 shows the comparison between real defect and optimized simulated images. As we can see, the location and the size of simulated pitting in optimized images are all consistent with the real defect. It is worth mentioning that the red artifact has disappeared, and bright spots have a big decline. The imaging clarity and accuracy are all improved. Meanwhile, the image of SDC has the least disturbing bright spots, whose definition is the highest. The image of E1 is the most symmetrical. Although there are still some disturbing bright spots compared with another two, it is better than before and does not affect the observation.

The following step is to judge the accuracy and clarity of optimized simulated images. The DRMS is calculated, and the simulated pitting images are zoomed in, respectively. According to Table 6, the maximum DRMS is only 0.1 mm. Compared with before, the position errors of SDC, RMSE and E1 are decreased by 100%, 55.4% and 50%, respectively. So, it is concluded that there is almost no position deviation that can ensure the position accuracy. As observed in Figure 11b–d, the sizes of simulated pitting in zoom-on images are all consistent with the real defect after optimization. We observed that the color of the red part is deepened and the outline is clearer. So, the clarity of imaging has been raised. It is worthy of mentioning that the zoom-in image of E1 has the best clarity and symmetry after amplification.

Finally, we draw a conclusion that the SPW-RAPID algorithm using characteristic parameters of SDC, RMSE and E1 can be used for locating and monitoring pitting corrosion on an aluminate alloy plate. The adjustment of corrosion characteristic parameters with sensor path weight can obtain better pitting images.

## 5. Conclusions

In this paper, the SPW-RAPID algorithm is proposed to monitor the pitting corrosion with a radius of 1 mm and a depth of 0.6 mm on a 2024-T3 aluminum alloy plate (400 mm × 400 m × 2 mm). According to characteristics of defect and specimen, a method for manufacturing pitting corrosion with ideal size is used. A square sensor network including 12 PZTs is designed. A ScanGenie-II integrated structural health monitoring scanning system is built to collect both the health signal and damage signal. The results are as follows:The A0 mode of Lamb wave is extracted to obtain corrosion damage characteristic information. The corrosion damage characteristic parameters including SDC, RMSE, E1, E2, E3 and NCM using the conventional RAPID algorithm is explored successively for imaging pitting corrosion. The results showed that the DRMS of the images based on SDC, RMSE, and E1 is up to 0.224 mm, while the DRMS of the images based on E2, E3 and NCM is up to 60.000 mm. So, it is found that only SDC, RMSE and E1 are sensitive to the small signal difference generated by pitting corrosion, and they can roughly locate the position of pitting corrosion. They were chosen as characteristic parameters for the SPW-RAPID algorithm;The SPW-RAPID algorithm based on the sensor path weight (*w*) is proposed, and its monitoring effect is better than the conventional RAPID algorithm. The value coefficient (*a*) is proposed according to the value of damage feature information carried by signals on the sensor path, and the influence factor (*l*) is put forward to evaluate the influence of the length of the sensor path on the extraction of pitting feature information. Finally, it is demonstrated that the sensor path weight (*w*) can be used to modify the corrosion damage characteristic parameters;The image simulated results obtained by the SPW-RAPID algorithm based on SDC, RMSE and E1 are all improved. As for image accuracy, the DRMS of images based on SDC, RMSE and E1 is 0 mm, 0.1 mm, 0.1 mm, respectively. It is revealed that the position errors of SDC, RMSE and E1 are decreased by 100%, 55.4% and 50%, respectively. With regard to image clarity, the imaging area of SDC, RMSE and E1 is increased and clearer. In summary, the proposed SPW-RAPID algorithm using SDC, RMSE and E1 can locate the position of pitting corrosion with a radius of 1 mm and a depth of 0.6 mm.

## Figures and Tables

**Figure 1 materials-15-03887-f001:**
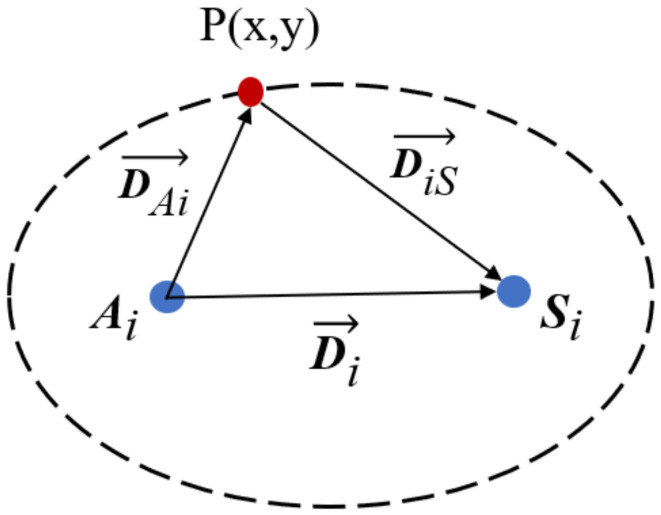
Illustration of RAPID algorithm.

**Figure 2 materials-15-03887-f002:**
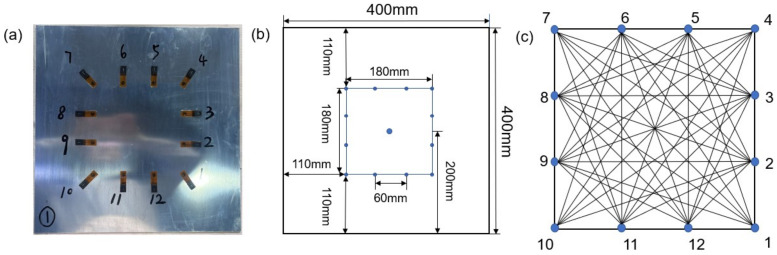
Designs (**a**) Test specimen. (**b**) Specimen diagram. (**c**) The layout of piezoelectric transducers network.

**Figure 3 materials-15-03887-f003:**
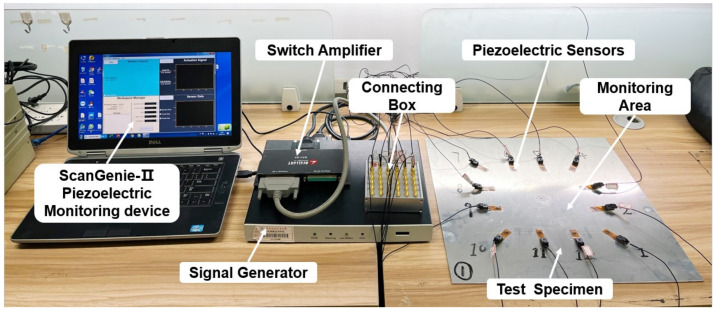
The experiment setup.

**Figure 4 materials-15-03887-f004:**
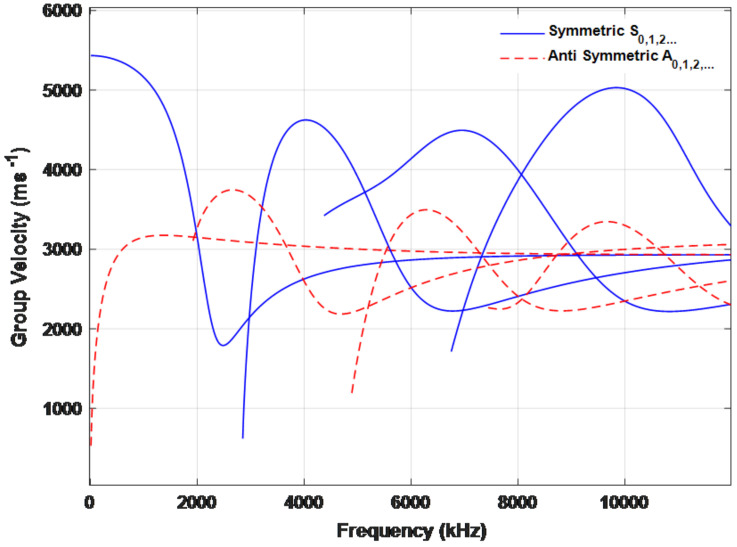
Dispersion curve of the Lamb wave of a 2 mm thick aluminum plate.

**Figure 5 materials-15-03887-f005:**
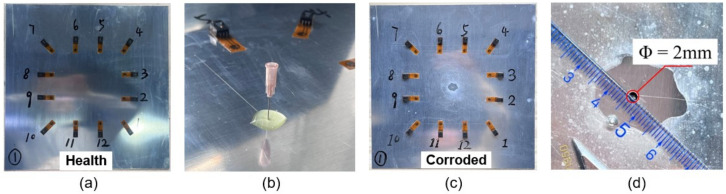
The manufacture process: (**a**) Undamaged specimen. (**b**) Corrosion process. (**c**) Damaged specimen. (**d**) Enlarged view of simulated pitting corrosion.

**Figure 6 materials-15-03887-f006:**
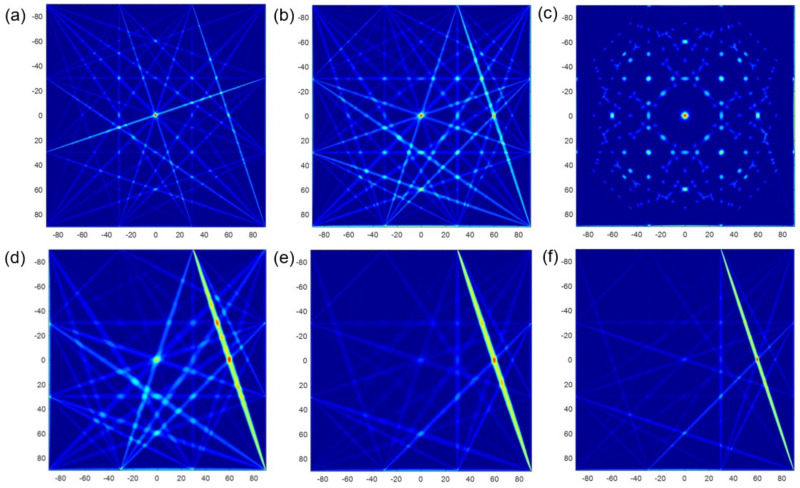
Simulated images based on different corrosion characteristic parameters: (**a**) SDC. (**b**) RMSE. (**c**) *E*1. (**d**) *E*2. (**e**) *E*3. (**f**) NCM.

**Figure 7 materials-15-03887-f007:**
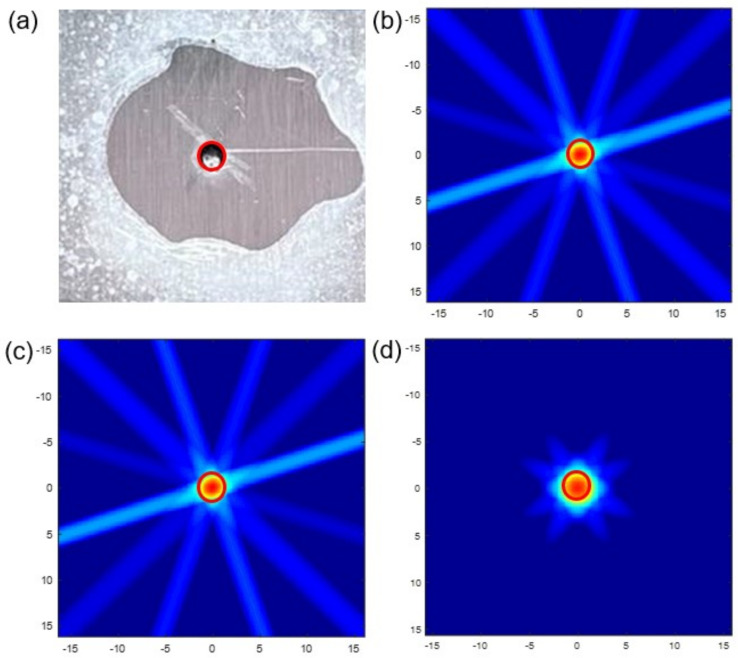
Enlarged view of real defect and simulated damage: (**a**) real defect. (**b**) SDC. (**c**) RMSE. (**d**) *E*1.

**Figure 8 materials-15-03887-f008:**
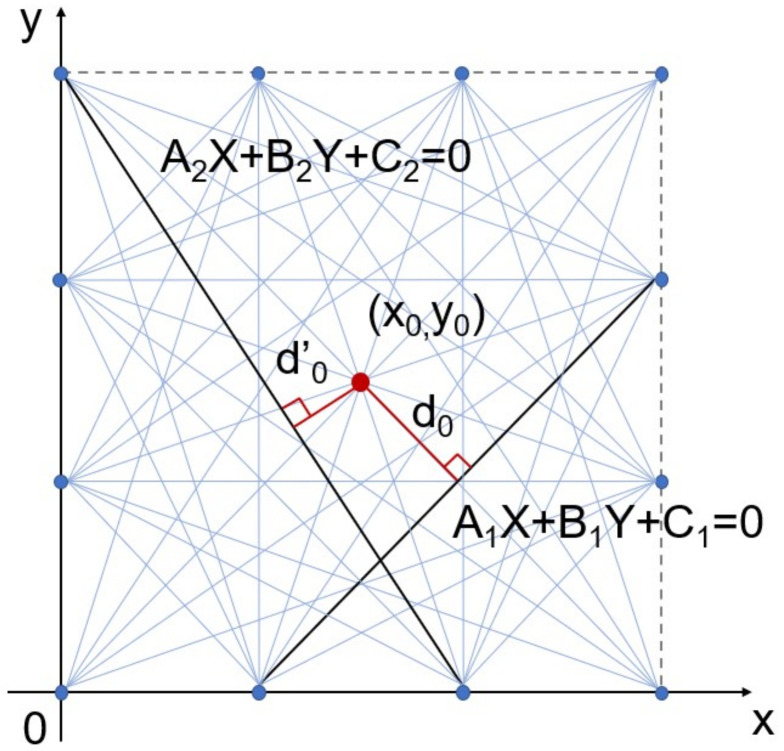
Rectangular coordinate system based on sensor network.

**Figure 9 materials-15-03887-f009:**
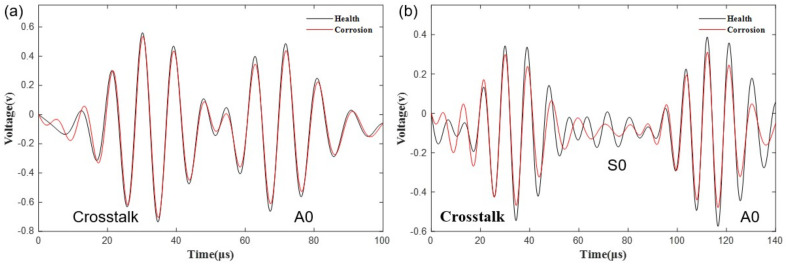
Comparison between health signal and damaged signal (**a**) on paths 3–5, (**b**) on paths 6–12.

**Figure 10 materials-15-03887-f010:**
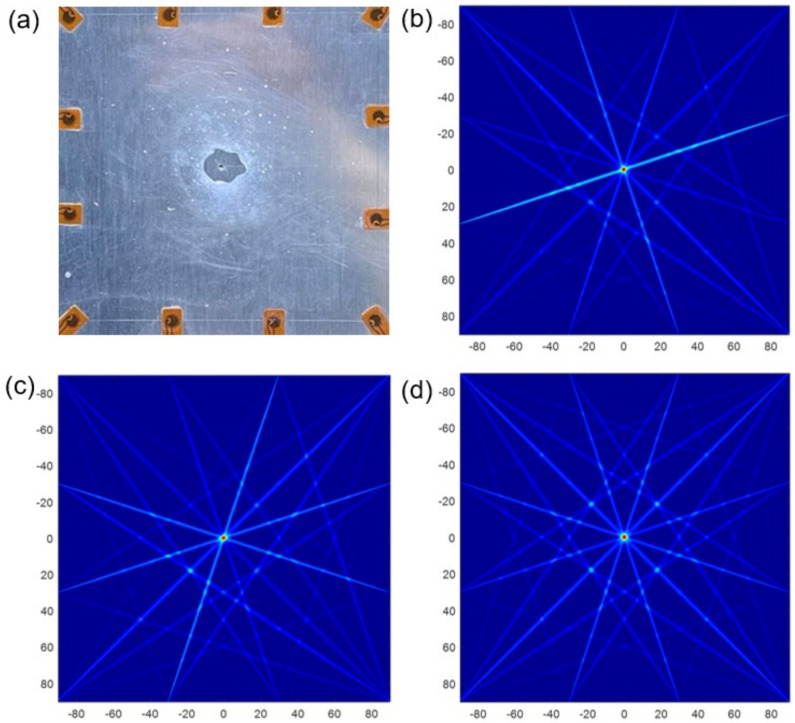
Comparison between real defect and simulated damage (**a**) real defect. (**b**) SDC. (**c**) RMSE. (**d**) *E*1.

**Figure 11 materials-15-03887-f011:**
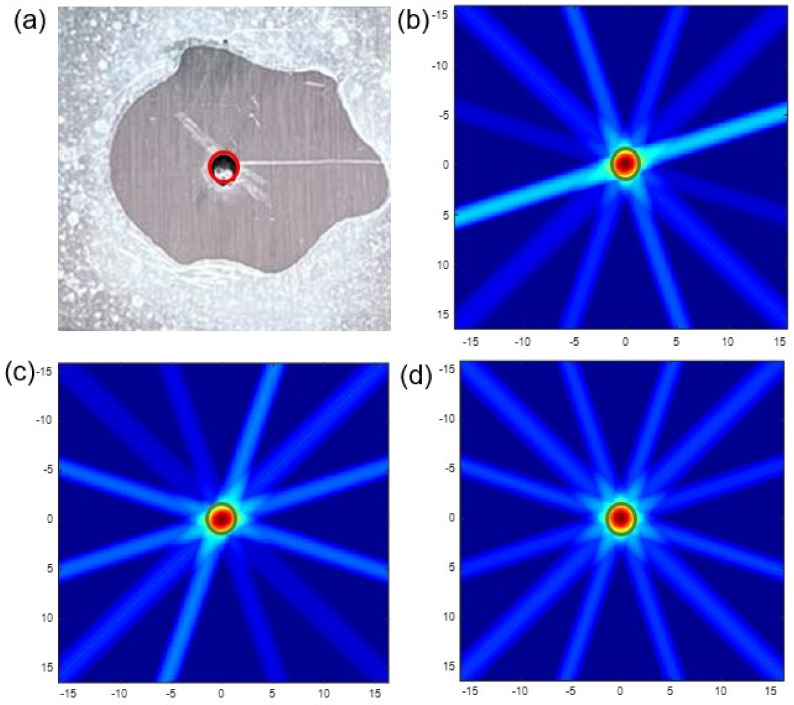
Enlarged views of real defect and simulated pitting corrosion (**a**) real defect. (**b**) SDC. (**c**) RMSE. (**d**) *E*1.

**Table 1 materials-15-03887-t001:** Material properties of specimen.

Specimen Material Grade	Aluminum 2024-T3
Poisson’s ratio	0.33
Density	2.78 g/cc
Young’s modulus	73.1 GPa
Fatigue strength	130 MPa

**Table 2 materials-15-03887-t002:** Piezoelectric monitoring equipment technical indicators.

Device Number	ScanGenie-II
Frequency	100–150 kHz
Conversion Rates	1.5, 6, 12, 24, 48 MHz
Integrated Power Amplifier	±50 V
Memory	16,000 Samples
Sampling Rates	130 MPa
Resolution	12-bit
ADC Range	±1 V

**Table 3 materials-15-03887-t003:** The chemical composition of hydrofluoric acid.

Content	Impurity Content (%)
Content of HF	≥40
Fe	≤0.0001
Cl	≤0.001
${\mathrm{PO}}_{4}$	≤0.0002
Heavy metal (Pb)	≤0.0005
Fluorosilicate (${\mathrm{SiF}}_{6}$)	≤0.04
Others	≤0.004

**Table 4 materials-15-03887-t004:** The original DRMS of real defect and simulated damage.

Parameters	$\left(x_{0},y_{0}\right)$	$\left(x_{i},y_{i}\right)$	$\sigma_{x}$/mm	$\sigma_{y}$/mm	DRMS/mm
SDC	(0, 0)	(−0.1, −0.1)	0.1	0.1	0.141
RMSE	(0, 0)	(−0.1, −0.2)	0.1	0.2	0.224
E1	(0, 0)	(0, −0.2)	0	0.2	0.200
E2	(0, 0)	(59.9, −0.2)	59.9	0.2	59.900
E3	(0, 0)	(60.0, −0.2)	60.0	0.2	60.000
NCM	(0, 0)	(59.9, −0.3)	59.9	0.3	59.901

**Table 5 materials-15-03887-t005:** The optimized DRMS of real defect and simulated damage.

$d_{0}$/cm	Number of Paths	*f*	$a_{ij}$
0.00	6	High value	1
2.50	8	Good value	0.5
3.00	4	General value	0.2
4.24	4	General value	0.2
5.69	8	General value	0.2
6.71	8	General value	0.2
8.49	4	General value	0.2
9.00	24	No value	0

**Table 6 materials-15-03887-t006:** Position error of real defect and simulated pitting.

Parameters	$\left(x_{0},y_{0}\right)$	$\left(x_{i},y_{i}\right)$	$\sigma_{x}$/mm	$\sigma_{y}$/mm	DRMS/mm
SDC	(0, 0)	(0, 0)	0	0	0
RMSE	(0, 0)	(0, −0.1)	0	0.1	0.1
E1	(0, 0)	(0, −0.1)	0	0.1	0.1

## Data Availability

The data presented in the current study are available upon request from the corresponding author.

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
