# Peer review of "Application of Sensor Path Weighting RAPID Algorithm on Pitting Corrosion Monitoring of Aluminum Plate"

_materials, 2022, doi:10.3390/ma15113887_

Round 1
Reviewer 1 Report
Very interesting article, it would be worth adding information what would happen if we used fiber optic sensors, e.g. FBG, instead of piezoelectric sensors.
The article is interesting, but it would be worth extending. Of course, if the authors do not decide to supplement this amendment, it is not disqualifying, but in the opinion of the reviewer it would extend the area of interest for the article.
Author Response
Dear editor:
Thank you for your letter and for the reviewer's comments concerning our manuscript entitled “Application of sensor path weighting RAPID algorithm on pitting corrosion monitoring of Aluminum plate” (ID: materials-1717597). Those comments are all valuable and very helpful for revising and improving our paper, as well as the important guiding significance to our researches. We have studied comments carefully and the main responds to the reviewer's comments are as following:
Response to comment: what would happen if we used fiber optic sensors, e.g. FBG, instead of piezoelectric sensors.
Response:
Firstly, it should be noticed that the sensor path weighting RAPID algorithm proposed in this paper is based on the characteristic parameters of Lamb wave A0 mode, which needs to be extracted and processed. When using the fiber optic sensors to monitor the damage, the signal generated by fiber optic sensors cannot be directly used, because it needs to be modulated. This process increases the complexity of the monitoring method.
Secondly, the modulation process of signal generated by fiber optic sensors is also difficult and inconvenient, which will cause big errors. Because the S0 mode and A0 mode wave packets may not be separated completely during this process. While the algorithm in this paper has high requirements for the integrity of wave modes, as we can find in the article at section 2.2.
Finally, the voltage of signal generated by fiber optic sensors is too small, although the signal amplifier is used, the difference of signal amplitude is still subtle. While the signal difference caused by the tiny pitting corrosion in this paper is small as well. If there is a large error during the signal excitation and acquisition process, it is not conducive to the subsequent imaging of pitting corrosion.
To sum up, the method proposed in this paper is not conducive to the application of fiber optic sensors in theory, and we have not carried out relevant experiments using fiber optic sensors to prove their effect and applicability.
Thank you and all the reviewers for the kind advice.
Sincerely yours.
Reviewer 2 Report
just a few notes
- Line 243. „a medical syringe with a needle diameter of 1mm is used to drill a small hole at the pre-etching area”. How can you drill a hole with a syringe on the metal?
- Moreover, how do you check the reproducibility of the pit? are all pits the same size?
- In the Discussion and results, the figures in the text are written wrongly. E.g. figure 7 (line 255) and figure 8 (277) in the text, refer to figures 6 and 7.
Author Response
Dear editor:
Thank you for your letter and for the reviewer's comments concerning our manuscript entitled “Application of sensor path weighting RAPID algorithm on pitting corrosion monitoring of Aluminum plate” (ID: materials-1717597). Those comments are all valuable and very helpful for revising and improving our paper, as well as the important guiding significance to our researches. We have studied comments carefully and the main responds to the reviewer's comments are as following:
Response to comment 1: Line 243. „a medical syringe with a needle diameter of 1mm is used to drill a small hole at the pre-etching area”. How can you drill a hole with a syringe on the metal?
Response: “a small hole” was made on the glass adhesive layer above the pre-etching area with the needle of syringe, not on the metal. I will explain it more clearly in the text.
Response to comment 2: Moreover, how do you check the reproducibility of the pit? are all pits the same size?
Response: the size and depth of pitting corrosion is controlled by the size of simulated hole on glass adhesive layer and the corrosion time.
Before the glass adhesive is completely cured, the simulated holes are all made by the needles with a diameter of 1mm and we also control the size of simulated holes during the process of manufacturing them.
Besides, we applied the corrosive solution with the same composition and concentration, and control the contact time between the metal and corrosive solution to ensure the same hole size.
After the damage is made, we will measure it with vernier caliper to check the reproducibility of the pits.
Response to comment 3: In the Discussion and results, the figures in the text are written wrongly. E.g. figure 7 (line 255) and figure 8 (277) in the text, refer to figures 6 and 7.
Response: we have corrected this mistake in the text.
Thank you and all the reviewers for the kind advice.
Sincerely yours.
Reviewer 3 Report
The article on the topic "Application of sensor path weighting RAPID algorithm on pitting corrosion monitoring of Aluminum plate" has novelty and scientific potential. The research work could be useful for practical applications as it provides an available way to monitoring tiny corrosion damage on aluminum alloy structure. In my opinion, the work can be published, but after the following corrections:
- There are not enough keywords, so expand the "characteristic parameters" by 1-3 keywords.
- At the beginning of the article, there is clearly not enough list of abbreviations that the authors use in the article.
- It is necessary to bring the design of references to literary sources to one form. It is correct if you put a space before each link in the text (the text of the article... space [source number]).
- It is more correct to indicate references to formulas in parentheses directly in the text describing them, i.e. as well as literary sources. But it concerns formulas 1-8 more. Also on formulas 9-15 and 17, reference numbers are not given in the text.
- In the sentence on lines 249-250 "In the end, a pitting corrosion with a radius of 1mm and a of 0.6mm is produced, as shown in Figure 5 (c) and (d)." should begin "As a result of this etching of an aluminum plate, a single pitting with a radius 1±… mm and a depth 0.6±… mm is obtained...".

Author Response
Dear editor:
Thank you for your letter and for the reviewer's comments concerning our manuscript entitled “Application of sensor path weighting RAPID algorithm on pitting corrosion monitoring of Aluminum plate” (ID: materials-1717597). Those comments are all valuable and very helpful for revising and improving our paper, as well as the important guiding significance to our researches. We have studied comments carefully and the main responds to the reviewer's comments are as following:
Response to comment 1: There are not enough keywords, so expand the "characteristic parameters" by 1-3 keywords.
Response: we have expanded “characteristic parameters” to “SDC”, “RMSE” and “E1” in the text.
Response to comment 2: At the beginning of the article, there is clearly not enough list of abbreviations that the authors use in the article.
Response: we have marked all the abbreviations in the text when they appeared at the first time, and adopted the abbreviation form when they appeared at the second time.
Response to comment 3: It is necessary to bring the design of references to literary sources to one form. It is correct if you put a space before each link in the text (the text of the article... space [source number]).
Response: we have corrected those mistakes in the text.
Response to comment 4: It is more correct to indicate references to formulas in parentheses directly in the text describing them, i.e. as well as literary sources. But it concerns formulas 1-8 more. Also on formulas 9-15 and 17, reference numbers are not given in the text.
Response: we have added the references of formulas 1-11 and 14-17 in the text. While the formulas 12-13 and 15-16 is proposed by ourselves and there are no references.
Response to comment 5: In the sentence on lines 249-250 "In the end, a pitting corrosion with a radius of 1mm and a of 0.6mm is produced, as shown in Figure 5 (c) and (d)." should begin "As a result of this etching of an aluminum plate, a single pitting with a radius 1±… mm and a depth 0.6±… mm is obtained...".
Response: we have corrected this content in the text.
Thank you and all the reviewers for the kind advice.
Sincerely yours.